# Using Mobile Edge AI to Detect and Map Diseases in Citrus Orchards

**DOI:** 10.3390/s23042165

**Published:** 2023-02-14

**Authors:** Jonathan C. F. da Silva, Mateus Coelho Silva, Eduardo J. S. Luz, Saul Delabrida, Ricardo A. R. Oliveira

**Affiliations:** Departamento de Computação, Instituto de Ciências Exatas e Biológicas, Universidade Federal de Ouro Preto, Rua Diogo Vasconcelos-128-Bauxita, Ouro Preto 35400-000, MG, Brazil

**Keywords:** edge AI, mobile edge computing, deep learning, citrus orchards

## Abstract

Deep Learning models have presented promising results when applied to Agriculture 4.0. Among other applications, these models can be used in disease detection and fruit counting. Deep Learning models usually have many layers in the architecture and millions of parameters. This aspect hinders the use of Deep Learning on mobile devices as they require a large amount of processing power for inference. In addition, the lack of high-quality Internet connectivity in the field impedes the usage of cloud computing, pushing the processing towards edge devices. This work describes the proposal of an edge AI application to detect and map diseases in citrus orchards. The proposed system has low computational demand, enabling the use of low-footprint models for both detection and classification tasks. We initially compared AI algorithms to detect fruits on trees. Specifically, we analyzed and compared YOLO and Faster R-CNN. Then, we studied lean AI models to perform the classification task. In this context, we tested and compared the performance of MobileNetV2, EfficientNetV2-B0, and NASNet-Mobile. In the detection task, YOLO and Faster R-CNN had similar AI performance metrics, but YOLO was significantly faster. In the image classification task, MobileNetMobileV2 and EfficientNetV2-B0 obtained an accuracy of 100%, while NASNet-Mobile had a 98% performance. As for the timing performance, MobileNetV2 and EfficientNetV2-B0 were the best candidates, while NASNet-Mobile was significantly worse. Furthermore, MobileNetV2 had a 10% better performance than EfficientNetV2-B0. Finally, we provide a method to evaluate the results from these algorithms towards describing the disease spread using statistical parametric models and a genetic algorithm to perform the parameters’ regression. With these results, we validated the proposed pipeline, enabling the usage of adequate AI models to develop a mobile edge AI solution.

## 1. Introduction

Deep Learning (DL) algorithms are increasingly embedded in agriculture [1]. This sector can benefit from these techniques for more modern, economical, and safe processes. In this context, DL models associated with edge systems become a tool that enables technological advances in the area, such as in citrus cultivation. For instance, it helps detect diseases through images collected in the environment [2].

The orange is one of the most-cultivated fruits in the world and generates a GDP of USD 6.5 billion in all countries in the production chain [3,4]; this requires efficient cultivation of these fruits. A crucial aspect of improving the productivity of the orange crop is detecting diseases through inspection. For instance, estimated losses in citrus crops can reach up to 22% [5] due to black spots.

Among the primary diseases in orange groves, some main pests are black spots, citrus canker, and greening. These diseases reduce oranges’ quality, caused by fungi or bacteria contamination, reducing the production due to the premature falling of the fruit from the trees.

In addition, the market is restricted due to these factors, which make it challenging to produce [6]. Other issues related to these diseases are fruits with acidic and bitter flavors and a bad appearance on the surface. Thus, these products become unsuitable for commercialization, such as producing fresh fruits and juices [7]. Figure 1 shows an orange with contamination by black spot fungi.

Agriculture has benefited from tools that use DL models, as they optimize traditional production processes and reduce losses from attacks from pests and diseases [9], for example real-time weed spraying using a computer vision application [10,11].

This work proposes a pipeline that allows the creation of a mobile application as an integrated DL model for identifying diseases in citrus. For this, we review the DL models’ applications in the literature, such as their performance in image detection and classification task. Then, we propose a set of algorithms to detect citrus fruits in orchards, evaluate the presence of diseases on each fruit, and map the disease spread among the orchard area. Finally, we evaluated an algorithm to model the disease spread distribution in the orchard area. Using these algorithms in a device that does not require cloud services can assist citrus growers in cultivating citrus in the orchards. Thus, the objective of this text is to:Propose and evaluate a path for using mobile edge AI to create an application to recognize diseases and their spread pattern among orchards.

In this context, we expected to evaluate a complete set of tools to perform the disease detection and mapping within orchards. Hence, the main contributions of this work are as follows:An evaluation and comparison of Deep Learning algorithms to detect citrus fruits in an image. At this time, the authors have not found any other authors evaluating such solutions in this context;An evaluation and comparison of lightweight Deep Learning algorithms to classify fruits as healthy or diseased, within the context of three diseases. The authors also did not find other authors performing the same kind of study;A proposal of a method based on an evolutionary computing algorithm to employ the gathered data to generate knowledge about the disease distribution throughout the orchard area. This approach is also unprecedented, although it has theoretical support.

For this, we discuss the theoretical references and related work in Section 2. We provide the materials and methods used to evaluate this work in Section 3. The results of these evaluations and preliminary discussions are presented in Section 4. Finally, we assess the conclusions obtained from this work in Section 5.

## 2. Theoretical References and Related Work

This section presents some traditional DL models used for image classification and detection in agriculture. In addition, we provide some of the most-relevant related works and how they approach the topic and differ from our work.

In the literature, various works use DL for image classification, such as ResNet [12], Inception [13], VGG [14], MobileNets [15], and NASNet-Mobile [16]. Some models have thousands of parameters with very deep architectures. This aspect can be a significant factor in choosing the model for integration into the mobile application. That is because a mobile device has limited resources that need to be preserved. In the following sections, we compare the DL models and mobile applications developed in this context.

### 2.1. Analysis of Deep Learning Models in Agriculture

In an image classification task, with the plant village database with 38 different classes, including diseased and healthy images of leaves from 14 plants [17], the authors compared the models VGG 16, Inception V4, ResNet with 50, 101, and 152 layers, and DenseNet with 121 layers. They concluded that DenseNet consistently improved the accuracy with an increasing number of epochs, with no signs of overfitting and performance deterioration. In addition, DenseNet requires considerably fewer parameters and achieved an accuracy of 99.75%, surpassing the other models.

For classifying apple leaf diseases, for a dataset containing 3651 images and four categories, scab, healthy, multiple infections, and apple rust [18], the ResNet50 and VGG19 models reached 87.7% accuracy for the tests performed. In another literature work [19], for the detection of diseases in rice, ResNet101V2 was the best-performing model with an accuracy of 86.79%. This work compared the VGG16, VGG19, ResNet50, and ResNet50V2 models. The metrics most used in this literature to calculate the performance of DL models are the precision, recall, and F1-score.

Mobile devices require models with fewer parameters to preserve device resources and execution with low latency. On the one hand, they are ideal for detecting diseases using AI. On the other hand, most works do not evaluate the resource footprint considering the application on edge devices. This literature review shows how deep models have achieved desirable performances within the context of agricultural applications.

### 2.2. Mobile Applications Using Deep Learning in Agriculture

With Agriculture 4.0, intelligent applications are increasingly utilized to solve problems on plantations, such as disease identification [20]. In the literature, some works use DL models in this context, such as the one proposed by Chen et al. [21], which uses a YOLOv3 model on a smartphone to detect pests on plantations and obtained 90% accuracy in performing the task.

Thai-Nghe et al. [22] used an EfficientNet model integrated with a device, obtaining an accuracy of 95% and a response time of 1.7s. However, they did not analyze the application’s processing consumption, considering that this application may be in places where re-powering the device during the activity is not possible.

Verma et al. [23] proposed a mobile application for diagnosing plant diseases and compared some DL models: ResNetv2, VGG16, VGG19, ResNet50, and Xception. They chose ResNet50 as the most accurate to create the mobile application. Although this model selected by the authors achieved considerable accuracy for them, this model has significantly more parameters compared to MobileNetV2. This makes the model demand more resources for processing on the device.

An application for pest detection on plantations can face communication problems if the processing is performed in the cloud, as proposed by Karar et al. [24]. Contrary to the cited literature, our work suggests edge processing to avoid communication problems in the rating system.

Barman and Choudhury [25] designed a smartphone app to detect diseases on citrus leaves. They relied on CNNs to classify the samples based on the collected images. Their work is related to ours in aim and in part of the applications, but the authors needed to segment their leaves manually. Our process is non-intrusive and segments the orange images while in the orchard.

Pan et al. [26] also developed a smartphone-based solution to classify diseases in citrus fruits and on their leaves. They relied on a cloud-based solution to perform the classification task and returned its result to the user. Their results are relate to this work, but require the user to segment the image manually and require a cloud service. Our proposal is edge-based and performs the segmentation and classification in scale on its own.

## 3. Materials and Methods

This section discusses the methodology we propose to find and classify the oranges in orchards. For this, we split our work into three main stages. In the first one, the task was to find oranges in an image. The mobile application is capable of associating these samples with the geolocation. Then, we investigated two networks to infer the diseases from the citrus in the application. Finally, we evaluated how these inferences could be used to map diseases in an orchard.

All algorithmic tests were conducted on the same machine. It had an i5-9600K CPU and 32 GB of RAM. Furthermore, it had an NVidia GeForce RTX 2060 Super graphics card, supporting CUDA operations for machine learning acceleration. This machine was used to standardize the tests in both the detection and classification tasks.

### 3.1. Citrus Detection Methods: YOLO-v3 and Faster R-CNN

The first step in performing this was detecting the citrus fruits within the orchard. For this, we performed a case study to validate the method to sample the fruits in the orchard using the image. This step aimed to prove the concept and compare the performance of two AI detection solutions in this context. We initially chose YOLO-V3 and Faster R-CNN as the architectures to perform this task. These networks were chosen as they are Deep Learning alternatives for object detection in images. As this solution is unique to the best of our knowledge, these models can work as a benchmark for future implementations.

An object detection network is usually composed of two parts, a pre-trained ImageNet backbone and a head used to predict the classes and bounding boxes [27]. One of the most-representative models in this context is YOLO [28]. The standard network YOLO has 24 convolutional layers, followed by two fully connected layers. With this, YOLO predicts multiple bounding boxes. First, it splits an image into a grid of cells. Then, it predicts the bounding boxes by a threshold, according to the object’s position in the image [29].

In the literature, there are some works for detecting citrus with YOLO. Good accuracy was achieved by applying this method in edge computing, using the NVidia Jetson Xavier NX Hardware, in [30]. Other authors showed that YOLO-LITE ran at about 21 FPS on a non-GPU computer and at 10 FPS after being implemented on a website with only seven layers and 482 million FLOPS. This speed was 3.8-times faster than the fastest state-of-art-model, SSD MobilenetV1 [31]. Thus, we decided to use YOLO to implement the mobile devices for object detection, integrating a mobile application.

We initially established a dataset to work towards this goal. Our choice was to use a custom-made dataset. We created it by initially downloading images from the Internet through the Google Images search engine. Then, we used the open-access tool labelImg [32] to create the annotations for this custom dataset in the Pascal VOC format. Figure 2 displays how this process works. We annotated oranges in 120 images for training and 40 for validation. Although this initial number of images is not very large, it was enough to establish a proof-of-concept for later development stages.

Then, we experimented with YOLO-v3 and Faster R-CNN. We used the Keras [33] weights as the back-end candidates. In this case, we explored two metrics. As the first one, we evaluated the mean average precision (mAP) in the object detection context. This metric considers the average percentage of correct predictions by varying the threshold of the accepted answers based on the intersection over union (IoU). In this case, we are detecting a single class. Thus, the mAP will be the same as the average precision calculated, varying the threshold for the accepted predictions. Figure 3 displays how this metric is calculated.

We also evaluated the advantage of using one model against another regarding the timing aspects. For this, we performed 30 rounds of predictions over eight images that did not belong to the dataset. We tested the models using both back-ends for the same images, evaluating the average time to perform the predictions for each image. For this, we performed a *t*-test analysis for each of the eight images. In these tests, the hypotheses were as follows:**$H_{0}$:** The two samplings have no significant timing difference for both models.**$H_{1}$:** The average times to perform the predictions are different for each model.

We know that if the values of the mAP are similar, the timing can be a constraint in deciding to use one or the other model. Nonetheless, if there is no significant difference in the timing, we can choose the model with the best mAP.

### 3.2. Citrus Classification Methods: MobileNet-V2, EfficientNetV2-B0, and NASNet-Mobile

In this section, we study the options for the classification algorithms for citrus images. In our context, it is interesting to have models that may have a feasible performance in the embedded environment. Thus, we evaluated three different models that can solve the issue. From the available solutions in Keras [33], we selected three different models to work as backbones to solve this issue:**MobileNet-V2**: This model is a convolutional neural network (CNN) commonly used to solve classification problems [34]. Without its top layer, this model is 14 MB.**EfficientNetV2-B0**: This model is also a CNN commonly used for classification problems [35]. It is 29 MB in size without its top layers.**NASNet-Mobile**: This is another lightweight CNN used for classification problems [36]. This model is 23 MB without its top layers.

MobileNets are different from traditional convolutional networks, being small and fast [15]. Smaller MobileNets are faster as they use a width and resolution multiplier, trading a fair amount of precision to reduce the size and latency [15]. Thus, this DL model becomes a powerful option for integration into mobile applications, such as in image detection tasks such as for fruit diseases [37].

The NASNet architecture’s background the Neural Architecture Search (NAS) framework [38]. This Deep Learning (DL) model is flexible and scalable for different applications. In object detection, a miniature version of NASNet also achieved 74% top-1 accuracy, equivalent to state-of-the-art models for mobile platforms [16].

EfficientNets are a family of convolutional neural network models designed to be faster and more precise on classification tasks [39]. The most miniaturized model in this set is EfficientNet B0. It has been used in tasks such as waste classification [40] due to its efficiency and performance.

All three models were used to obtain a better classification potential considering the dataset used in Silva et al. [41]. The structure of the prediction model starts with a normalization layer for a three-channel image. Then, the data flow through the model backbone. Finally, the output of the backbone is flattened and goes through a dense layer with 512 neurons. The output is a dense layer with a “softmax” activation function. Figure 4 illustrates how we employed these backbones to build our model.

To evaluate these networks, we employed two different evaluations. Initially, we evaluated the models’ performance according to traditional machine learning metrics: precision, recall, F1-score, and global average. Then, we studied how each model performed regarding its timing constraints. As these predictions were performed on images of the same size, we tested their capability over the whole test dataset, measuring the average prediction time for each model. Then, we compared each pair of models with a *t*-test analysis. In these tests, the hypotheses were:**$H_{0}$:** The two samplings have no significant timing difference for both models.**$H_{1}$:** The average times to perform the predictions are different for each model.

If the models performed very similarly regarding their machine learning metrics, the timing constraint can again be a constraint in the model choice. With these tests and the previous versions, we could perform the complete detection and classification tasks. In the following subsection, we discuss how to use these data to evaluate the disease spread in an orchard.

### 3.3. Mapping Diseases in Orchards

Once having defined the algorithms used in the mobile tool, the following step was to evaluate how to use them in a real-world context. We started from the standpoint that disease detection is a critical task within a citrus orchard. Nonetheless, we also wanted to produce a further result with these data. Using ground-based measurements with GPS data, we expected not only to detect the diseases within the orchards, but to understand how they were affecting the orchard. Thus, comprehending these diseases’ spatial distribution is a valuable task within this context. Many works in the literature describe the spread of diseases in orchards [42,43,44,45,46], a fact that supported our decision.

In this work, we considered that each disease happens independently. To our knowledge, we have not found authors studying the presence of multiple diseases in the same orchard and how they mutually influence each other. Thus, we considered infections as isolated events in this initial approach.

We considered the possibility of ground-based sampling. As supported by some previous studies [41,47,48], this sampling can be supported by wearable computing solutions. Thus, we considered that a user wearing such solutions can perform these measurements in the field. We considered that these solutions were also paired with GPS data.

We know that as we provide faster information about diseases using AI, we can also run algorithms to approach this issue faster. We divided this into a few theoretical and experimental steps:Understanding the probabilistic distribution models that describe the spatial distribution of diseases in orchards;Selecting a realistic map to represent an “ideal” citrus orchard;Generating samples according to a parametric probabilistic model using a version of the Monte Carlo method;Performing regressions using various techniques to approach the initial model with various statistical samplings simulated using the previous method.

The first step towards this goal was to create a probabilistic spatial distribution model to describe how diseases spread in orchards. At a later stage, we could evaluate the dynamics of this spread, but this validation application will provide a picture of the orchard’s current status.

Costa et al. [45] studied the distribution of *Diaphorina citri* in citrus orchards through statistical tools. After testing the data for randomness, the authors employed a negative binomial distribution analysis to study the probabilities of finding this pest in the orchards. In their work, this distribution had a higher coefficient of determination when compared to the Poisson distribution.

In the proposal of Charest et al. [43], they evaluated the spatial distribution of a disease in an apple orchard by dividing the area into blocks. Then, they calculated the probability of finding disease spores for each block according to a Poisson distribution and a binomial distribution. A binomial distribution is a discrete probability representation that behaves similarly to normally distributed events sampled within a discrete number of classes.

As we observe a space according to its coordinates, our modeling works with a probability within a two-variable space. Thus, the probability for each event in a continuous bivariate space can be represented similarly to a bivariate normal Probability Function (PF), represented by Equation (Equation 1):(1)$$P\left(x,y\right)=D_{0}e^{\left(-\frac{1}{2(1-\rho^{2})}\left[{\left(\frac{x-\mu_{x}}{\sigma_{x}}\right)}^{2}-2\rho \left(\frac{x-\mu_{x}}{\sigma_{x}}\right)\left(\frac{y-\mu_{y}}{\sigma_{y}}\right)+{\left(\frac{y-\mu_{y}}{\sigma_{y}}\right)}^{2}\right]\right)}$$

The standard deviations parameterize the above equation among the *x* and *y* axes ($\sigma_{x}$,$\sigma_{y}$), the correlation between the two variables (ρ), the mean value on each axis ($\mu_{x}$,$\mu_{y}$), and the maximum disease density ($D_{0}$). These parameters allow determining the disease distribution $D_{0}$ according only to its spatial coordinates x,y, using a Gaussian parametric model.

The value of this maximum density can reach up to even 50% of the fruits [49]. It represents a probability function instead of a probability density function, as it is normalized by the maximum density of diseased fruits observed in the orchard. We can initially work starting from data with zero correlation for simplification purposes. Thus, the previous equation was simplified to Equation (Equation 2), representing non-correlated coordinates:(2)$$P\left(x,y\right)=D_{0}e^{\left(-\frac{1}{2}\left[{\left(\frac{x-\mu_{x}}{\sigma_{x}}\right)}^{2}+{\left(\frac{y-\mu_{y}}{\sigma_{y}}\right)}^{2}\right]\right)}$$

Another important aspect is to define an experimental area’s configuration. As such a dataset does not exist, we based our solution on a simulated environment based on real aspects of orchards. For instance, Marin et al. [50] studied a citrus orchard’s area of circa 0.63 ha (63 m × 100 m). Osco et al [51] evaluated a larger area, of 70.4 ha. As is shown, the range of these areas can vary. Our hypothetical citrus orchard sampling area consisted of a 1 ha square. The tree spacing followed a distance similar to that presented by Petillo et al. [52], 4 m × 6 m. This configuration’s sampling area consisted of 425 trees, as illustrated in Figure 5.

We can obtain different results when applying the probability function described in Equation (Equation 2) in the mentioned area with arbitrary parameters. We illustrate some of the options for these results in Figure 6. As we can see from the images, the five parameters of the parametric model ($D_{0}$,$\sigma_{x}$,$\sigma_{y}$,$\mu_{x}$,$\mu_{y}$) made significant alterations to the shapes and values of the probability function. The complete model, displayed in Equation Equation 1, allows for further manipulation of the shape of the probability function according to the correlation between the two variables (ρ). In Figure 7, it is possible to see how this parameter shapes the curve according to its value in the bivariate normal probability function.

These functions display how we can develop probability functions to describe disease spread in the orchard sampling area. With five or six parameters, depending on the function, we can obtain several distributions corresponding to how the literature displays functions that model spatially distributed diseases. The next step should provide a representative scenario of the hypothetical orchard area. The sampling scenario’s creation was through the Monte Carlo method, from its features in developing draws from a probability function. For this, we followed the perspective of the method in a general manner. According to Harrison [53], there is no unique way of applying the Monte Carlo method, but there are general lines.

In general terms, it starts from a defined probability function. Then, the simulation generates multiple random inputs classified according to this probability function. Finally, according to the interest, the data processing happens after the sampling batch’s creation. We used this method to simulate a sampling method within the orchard. Our steps within the Monte Carlo method were as follows:**Defining a probability function:** We define two sets of parametric models. One uses the simplified bivariate probability function, while the other uses the complete one.**Sampling the data:** For each tree, we generated 100 samples. Given the tree (*x*,*y*) coordinate, the probability of the disease being present is given by P(x,y). The probability of detection from each disease was considered to be its recall ($P_{r}$). Thus, a draw *D* within a uniform distribution will generate the classification result. The sample was considered diseased if $D<=P\left(x,y\right)\times P_{r}$.**Compute the sampling data:** With these sampling data in hand, our objective was to perform a regression to obtain the parameters from the probability functions that generated the samples. We tested the sampling considering all three diseases described by the networks from Section 3.2. Furthermore, we performed the tests with both simplified and complete PFs. Thus, we performed six tests to evaluate the results we should expect from the sampling process with the proposed system.

The number of samples was chosen according to the literature analysis. Iglesias et al. [54] had citrus trees with more than 160 fruits on average. Ouma [55] already studied citrus with a maximum value of 1400 fruits per tree. Given this variety of possibilities, as the automatic sampling and classification, we know the possibility of sampling 100 fruits from each tree. We used a uniform distribution to simulate the sampling of diseased and healthy citrus, considering the probability function at the spot.

To perform the data computation task, we used the process proposed by Silva et al. [56], which uses an evolutionary algorithm to perform a regression from samples and obtain the disease distribution. We designed an algorithm and ran it to obtain the results in all six cases. As the models were parametric, we considered the solution as a part of a hyperspace $R^{n}$. Thus, to score the obtained solutions, we used the following equation using the Euclidean distance between the original parameters and the obtained ones. We followed the definitions below to create this metric:

**Definition** **1.**
*Let V be the original parameters’ vector.*


**Definition** **2.**
*Let $\hat{V}$ be the obtained parameters’ vector.*


**Definition** **3.**
*Let $v_{i}$ be the element on the i-th position of the vector V.*


**Definition** **4.**
*Let $\hat{v_{i}}$ be the element on the i-th position of the vector $\hat{V}$.*


The Euclidean distance from the vectors is defined by:(3)$$D\left(V,\hat{V}\right)=\sqrt{\sum_{i}{\left(v_{i}-\hat{v_{1}}\right)}^{2}}$$

The score will be a factor calculated by:(4)$$S\left(V,\hat{V}\right)=\frac{1}{1.03^{D(V,\hat{V})}}$$

This score is helpful, as when the values of *V* and $\hat{V}$ are very similar, it comes closer to 1. Nonetheless, as it goes further away, the value asymptotically approaches 0 as the distance increases. The denominator is thought to dampen the score decay as the distance increases. Figure 8 illustrates this scoring system.

## 4. Results

In this section, we give the results obtained from the methods described in the previous sections. Initially, we give the results of running the YOLO and Faster R-CNN algorithms to detect fruits among the tree. Then, we compare the three proposed architectures to solve the classification issue, which were MobileNet-V2, EfficientNetV2-B0, and NASNet-Mobile. Finally, we evaluate the expected behavior of this application in the field using the proposed methodological framework.

### 4.1. Citrus Detection Method: YOLO-V3 and Faster R-CNN

For the citrus detection system, we trained two different implementations: YOLO-V3 and Faster R-CNN. As we showed in Section 3.1, the first step was creating markings in the images to compose a dataset. Then, we trained YOLO [57] and Faster R-CNN [58] according to implementations over the Keras framework. Then, we trained the network to obtain the average precision according to the selected data. Figure 9 displays the results obtained by each implementation.

Initially, the results of the mAP metric were similar. The YOLO-V3 achieved 88%, while Faster R-CNN achieved 90%. Although Faster R-CNN had a better performance indicator, the YOLO algorithm displayed better results visually. The results were very similar, bringing the timing aspect as a crucial constraint in evaluating and comparing the algorithms. As these results were similar, we analyzed this aspect also qualitatively.

We performed 30 predictions using both algorithms on the same set of images to test the timing constraints. Then, we evaluated the results through a *t*-test, as described in Section 3.1. Table 1 displays the obtained results. Initially, our test showed that the times were statistically different, rejecting the null hypothesis. Analyzing the time averages and standard deviations, the results displayed a better performance executing the detection through the YOLO algorithm.

This results are especially significant within the mobile context. The initial results indicated that using the YOLO algorithm is probably better within the mobile context, where computing power is limited. The precision of both algorithms was virtually the same. Nonetheless, from a qualitative view, YOLO might provide more information.

### 4.2. Citrus Classification Methods: MobileNet-V2, EfficientNetV2-B0, and NASNet-Mobile

In this set of tests, we evaluated the possibility of identifying the diseases in segmented citrus images. We also compared the performance of multiple candidate backbones within this context. This work aimed to provide a baseline to perform mobile citrus classification, as displayed in Figure 10. For this, we evaluated three models using different networks as the backbones for the application. Figure 4, displayed in Section 3.2, shows the organization of this model. We chose smaller networks capable of providing suitable solutions within the mobile context.

Initially, we trained the networks using the Keras API as the baseline to create our codes. We trained the networks using the categorical cross-entropy loss function and the Adam optimizer. As the test set was used only as an indicator, we employed it in the validation steps in this section. The training was set to stop when the loss from the training data reached a plateau. Figure 11 displays the results for the training sessions for each model. These results show no signals of overfitting, displaying a satisfactory convergence for each model.

Then, we evaluated the machine learning classification metrics. For each class, we evaluated the precision, recall, and F1-score. We also evaluated the global average for each case. Table 2 displays the metrics for MobileNetV2. Table 3 displays the metrics for EfficientNetV2-B0. Table 4 displays the metrics for NASNet-Mobile. The results indicated that NASNet-Mobile had a slightly worse performance when compared with the two other options. The metrics of MobileNetV2 and EfficientNetV2-B0 indicated they had the same performance on these examination metrics.

Our final evaluation in this step of the research was to analyze the timing constraints of each application. For this, we measured the time required to predict the classes of all images in the dataset. As the difference between each model was only the backbone, all timing differences were related to this aspect. Table 5 displays the results from this set of tests.

The results indicated a statistically significant difference between the times of each sampling set. The model with the best timing performance was MobileNetV2, while the worst was obtained with NASNet-Mobile. The relative difference between MobileNetV2 was circa 10%. As both MobileNetV2 and EfficientNetV2-B0 had the same performance considering the machine learning metrics, our result indicated that MobileNetV2 should perform better in the context of this application.

### 4.3. Mapping Diseases in Orchards

As we showed in Section 3.3, we started by defining arbitrary probability functions for each disease in this stage. As we discussed, in the initial stage, we started from the simplified equation considering a non-correlated distribution among the axes, represented by Equation (Equation 2). These equations are parametric models based on five parameters: the maximum density $D_{0}$, the mean value on the *x* and *y* axes ($\mu_{x}$,$\mu_{y}$), and the standard deviation among each axis ($\sigma_{x}$,$\sigma_{y}$). We individually evaluated the distributions of each disease, considering them to be independent. In Table 6, we display the parametric values used to obtain the PFs, which are displayed in Figure 12.

After this stage, we sampled 100 points for each tree, representing the sampling of the segmentation algorithm upon the tree. On each point, we used the bivariate probability as the criterion for the existence of a diseased fruit according to this function. Then, we used the recall value as the probability of correctly classifying the diseased fruit, given that it is a positive sample from the class. The number of diseased fruits in that tree represents the disease density at the given point.

We performed this regression with the help of an evolutionary algorithm. For each distribution, we performed a genetic algorithm that searched for the best fit for the arbitrary probability function that describes the density of diseased fruits per tree. Figure 13 displays the organization of this algorithm.

Initially, we started a population with random parameters. The genotype from this solution was a set of five or six integers, according to the presence or absence of the ρ parameter. The population size was a parameter from the algorithm and the number of offspring. Then, the algorithm eliminated the worst individuals and replaced them with the offspring of the remaining ones. Each gene of the offspring was the mean value between the parents’ genes. The new offspring can have mutations, which are minor changes in the values of their genes. After that, the algorithm allows a five-round local search for minor improvements in the value. Finally, the algorithm evaluates the population and finishes in case it meets the convergence criteria or reaches the maximum epochs. For this, the parameters of this algorithm were:**Population size:** 800 individuals;**Maximum number of epochs:** 1000 epochs;**Number of offspring:** 200 individuals;

The fitness function was calculated as the integral of the absolute error between the predicted density p(i) and the measured density d(i) at each *i* tree. As we studied discrete samples, it was measured as the sum of the errors among each sample, as presented in Equation (Equation 5):(5)$$IAE=\sum_{i}\left|d\left(i\right)-p\left(i\right)\right|$$

We tested the sampling generated using the described version of the Monte Carlo method. Our reference parameters were the ones presented in Table 6. Figure 14 displays the obtained results for each distribution. The upper row displays the distributions used to generate the sampling. The lower row displays the results after the regression using the genetic algorithm.

From the second perspective, we experimented with the function containing the correlation term ρ. For this, we employed the complete version of the probability function, given in Equation (Equation 1). Table 7 displays the parameters chosen for the following part of this experiment. Furthermore, Figure 15 displays the behavior of these distributions in space. As the figure shows, they slightly differed from the behaviors shown in Figure 12 due to the correlation term.

Again, we tested the sampling generated using the described version of the Monte Carlo method. Our reference parameters were the ones presented in Table 7. Figure 16 displays the obtained results for each distribution. In the other case, the upper row displays the distributions used to generate the sampling, and the lower row displays the results after the regression using the genetic algorithm.

In both experiments, we initially performed a qualitative analysis according to the images from the distributions. What Figure 14 shows is that the algorithm could approximate the original distributions with good performance. Figure 16 shows that the addition of the correlation factor ρ impaired the performance of the predictions, but they still could approximate several features from the distributions.

Then, we started with a quantitative analysis of the first experiment. Table 8 displays the results for each class in the first experiment. The score was obtained according to Equation (Equation 4), which is a radial gradient that diminishes as the solution goes further away from the original data. The scoring indicates that the approximations were suitable. Furthermore, the proximity of these values shows that the ranges probably can precisely describe the distribution of this disease throughout the orchard.

Finally, we also analyzed the results for the densities considering the existence of a correlation between the directions. For this, we added arbitrary values to the ρ parameter. Table 9 displays the results for each class in the second experiment. The scoring values indicated that the algorithm had a lower performance considering distributions marked by this correlation factor. We should expect that this technique is limited to distributions where the disease is distributed with low correlation among the coordinates. Nonetheless, the effect of this parameter in discovering the other configuration aspects of these distributions was minor, and it still gave a fair understanding of the disease spread’s geolocation and conditions.

With these results, we described how the proposed algorithms, methods, and applications could be employed in the field to describe the conditions of local citrus and the disease incidence in the orchard. Although the proposed technique has some limitations, it provides valuable information considering the disease’s geolocation and the spread’s dispersion.

## 5. Conclusions and Discussions

In this article, we proposed a complete pipeline that allows a mobile application to use AI processing at the edge to detect and map diseases in citrus orchards. This pipeline starts with the detection of citrus fruits throughout the orchard. Then, the application performs a classification of these images. Finally, the combination of AI data and geolocation allows an understanding of the distribution of these diseases in the orchard area.

Although some traditional models have achieved good performance, by reviewing the literature, we saw that a large number of parameters, for example the depth of the model, may require greater processing power for the AI when applied to mobile devices, which are limited in resources. Thus, we integrated a model that, despite not having the best accuracy among the deep models, reached an answer in a short time and did not need high-processing-power hardware.

Our initial evaluation showed that YOLO-V3 was suitable for detecting the citrus fruits within an image. This model outperformed Faster R-CNN regarding the timing constraints, with a similar result in the detection aspect. On average, this model can detect oranges in an orchard image in less than 100 ms. Thus, it is more adequate for creating a mobile application, as it is computationally restrained.

Then, we evaluated a set of classification algorithms. We observed that all algorithms had a global accuracy between 98% and 100% and concluded that MobileNetV2 has a good balance between accuracy and timing to perform this task in a mobile application. With a 10% worse timing performance, EfficientNetV2-B0 is still an eligible candidate to perform this task. Given the conditions, the performance of NASNet-Mobile was significantly worse.

Finally, we were able to map the disease’s spread considering parametric models sampling the disease spread within the orchard. We performed this task through a regression to a probability function based on the bivariate normal distribution. These results were better for more simplified models without spatial correlation among the coordinates within the disease.

This application can be used to optimize the cultivation of oranges, expanding the opportunity to visualize the spatial distribution of fruits affected by diseases quickly and accurately. Thus, the citrus grower can take some measures to manage the orange trees, such as spraying before the diseases spread throughout the orchard. Future works can explore this pipeline in a realistic context, applying the proposed technologies in orchards to identify their strengths and weaknesses.

The employment of this sensing technology can also evolve the solution into a risk-management tool for non-infected citrus. Users can evaluate the temporal dynamics of disease spread in an orchard. With such information, the measurement of a stationary condition can support the orchard management in taking more efficient measures to stop the disease’s spread.

## Figures and Tables

**Figure 1 sensors-23-02165-f001:**
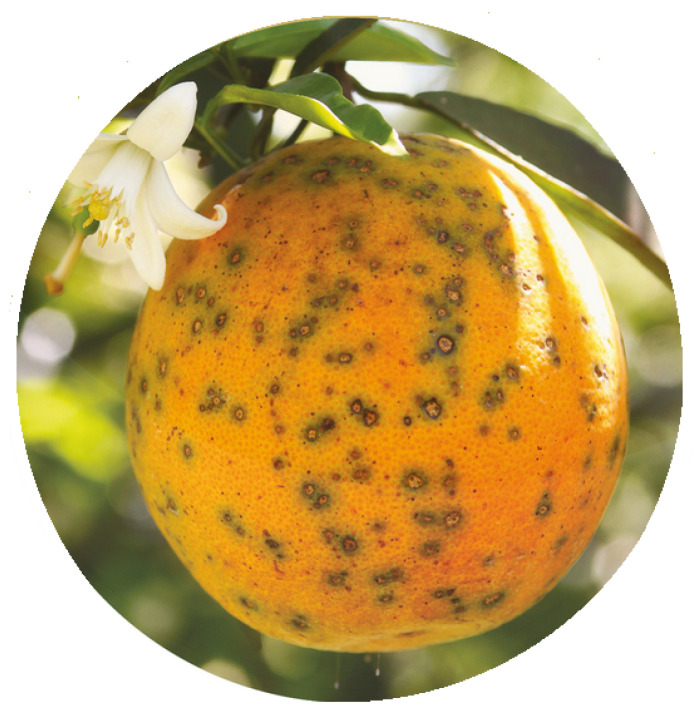
An example of black spot disease in an orange. Source: [8].

**Figure 2 sensors-23-02165-f002:**
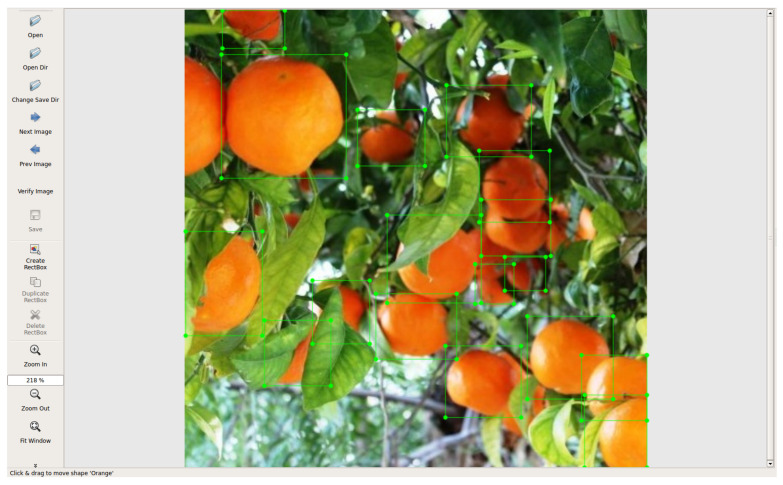
Images obtained from the annotation system.

**Figure 3 sensors-23-02165-f003:**
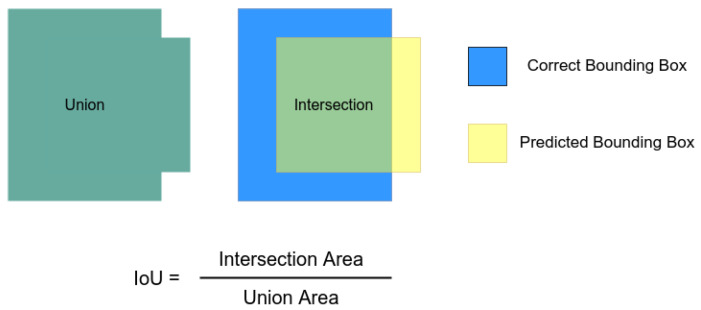
Intersection over union illustration.

**Figure 4 sensors-23-02165-f004:**
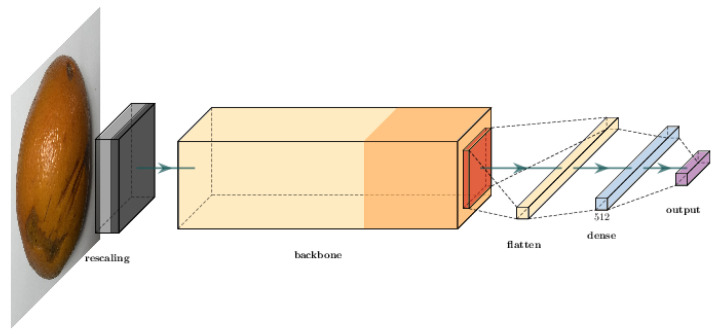
Classification network structure illustration.

**Figure 5 sensors-23-02165-f005:**
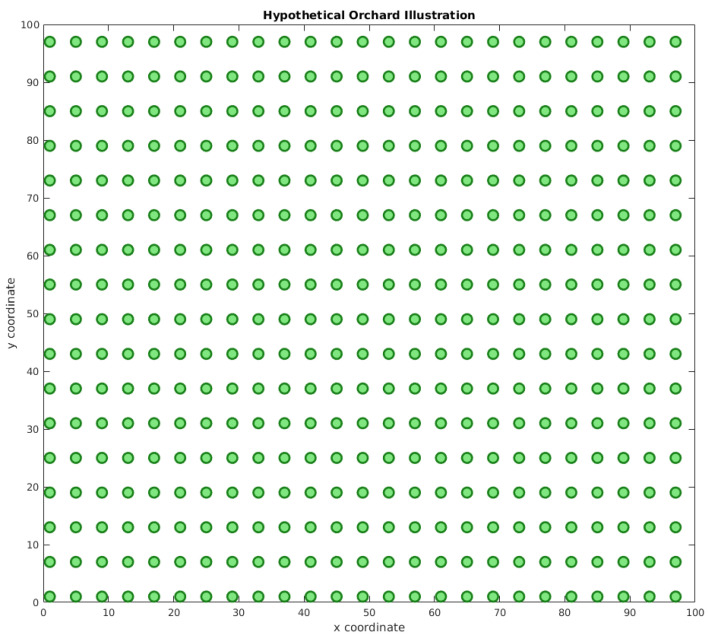
Hypothetical orchard illustration. The hypothetical orchard has 425 trees within a 1 ha area, separated into a 4 m × 6 m grid. The x and y coordinates represent the distance in meters.

**Figure 6 sensors-23-02165-f006:**
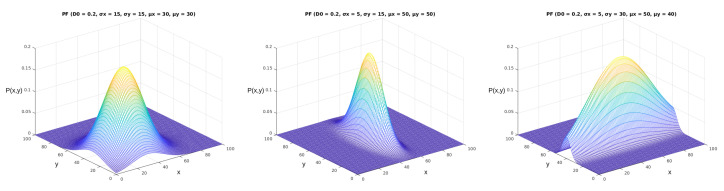
Simplified parameterized probability functions’ plot. These figures were generated using the function from Equation (Equation 2). The *x* and *y* axes represent the same coordinates represented in Figure 5, and the P(x,y) axis represents the values obtained from Equation (Equation 2).

**Figure 7 sensors-23-02165-f007:**
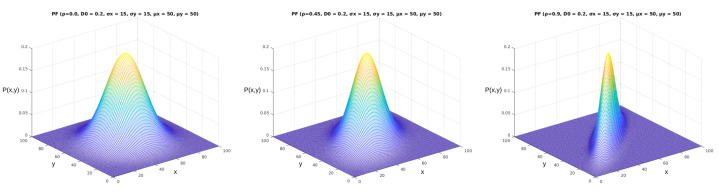
Complete parameterized probability functions plot. These figures were generated using the function from Equation (Equation 1). The *x* and *y* axes represent the same coordinates represented in Figure 5, and the P(x,y) axis represents the values obtained from Equation (Equation 1). In this figure, we can observe the effect of the ρ parameter, varying from 0 to 0.9.

**Figure 8 sensors-23-02165-f008:**
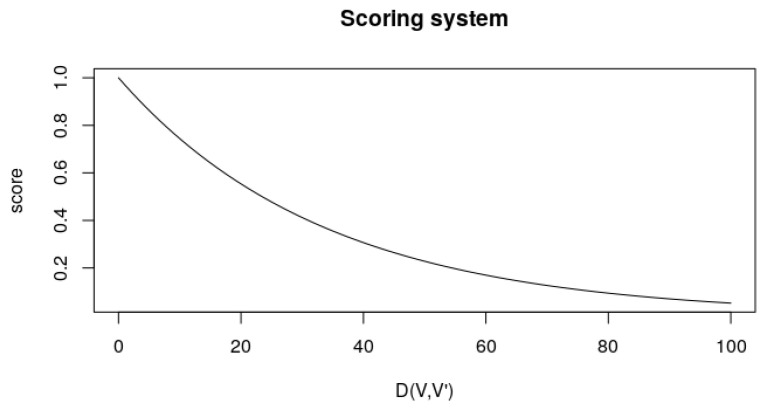
Scoring system according to Equation (Equation 4). The $D(V,V^{\prime })$ axis represents the distance between the ideal and obtained parameters, and the *score* axis represents the output from $S(V,\hat{V})$.

**Figure 9 sensors-23-02165-f009:**
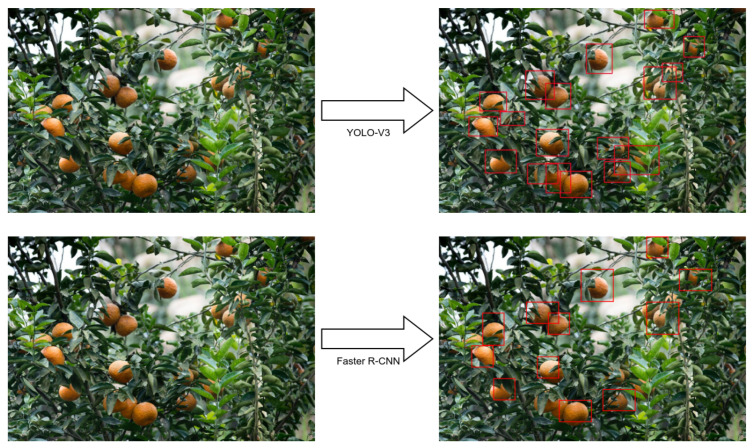
Result of the YOLO-V3 and Faster R-CNN training for this application.

**Figure 10 sensors-23-02165-f010:**
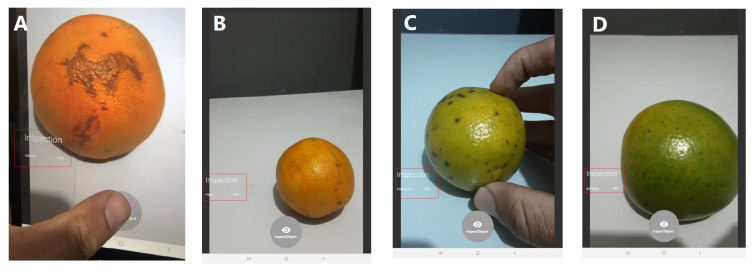
Detection of orange diseases with AI on the mobile device. Using a mobile application, the method could detect Citrus Canker (**A**), Fresh Oranges (**B**), Black Spot (**C**), and Greening (**D**).

**Figure 11 sensors-23-02165-f011:**
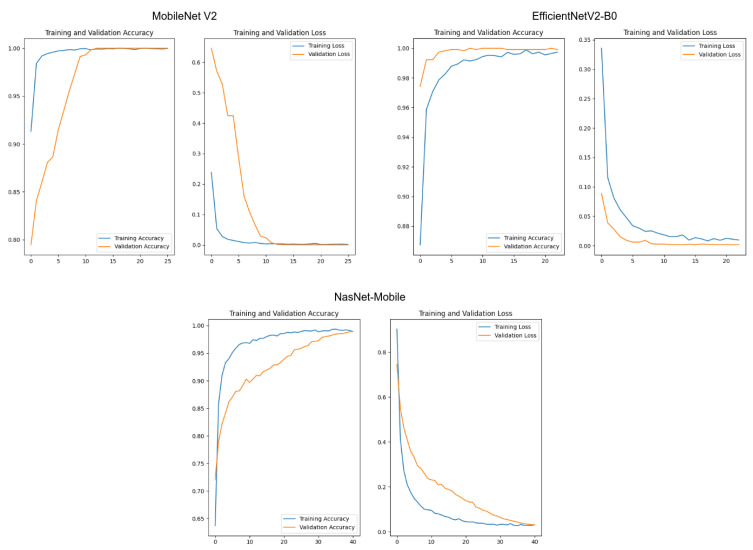
Evaluation of the accuracy and loss values for the training and validation sets. In all cases, the horizontal axes represent the training epochs. The vertical axes represent the accuracy or the loss function value, according to each title label.

**Figure 12 sensors-23-02165-f012:**
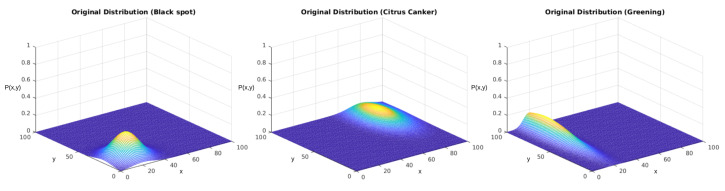
PFs displayed in Table 6 for each studied disease. The *x* and *y* axes represent the same coordinates represented in Figure 5, and the P(x,y) axis represents the values obtained from Equation (Equation 2).

**Figure 13 sensors-23-02165-f013:**
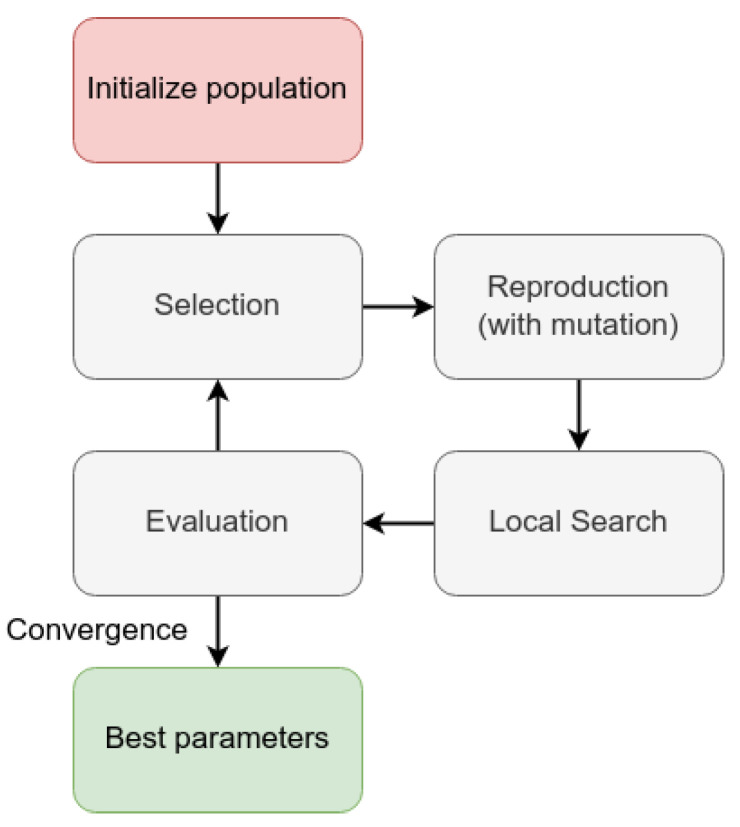
Evolutive algorithm organization.

**Figure 14 sensors-23-02165-f014:**
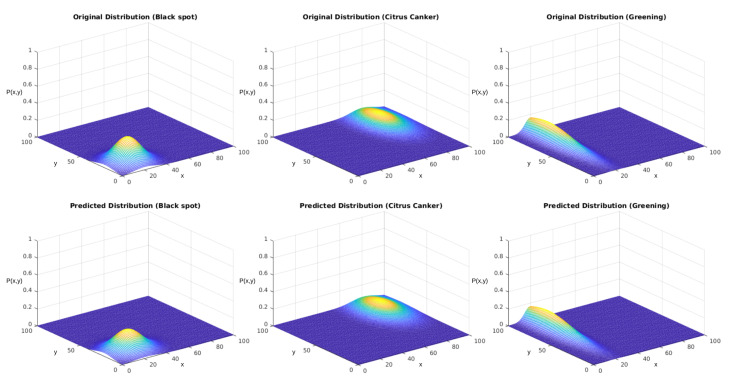
Predictions of the simplified PF using the evolutionary algorithm. The *x* and *y* axes represent the same coordinates represented in Figure 5, and the P(x,y) axis represents the values obtained from Equation (Equation 2).

**Figure 15 sensors-23-02165-f015:**
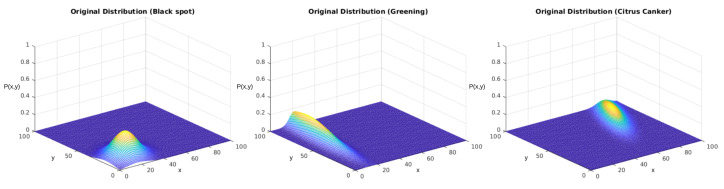
PFs displayed in Table 7 for each studied disease. The *x* and *y* axes represent the same coordinates represented in Figure 5, and the P(x,y) axis represents the values obtained from Equation (Equation 1).

**Figure 16 sensors-23-02165-f016:**
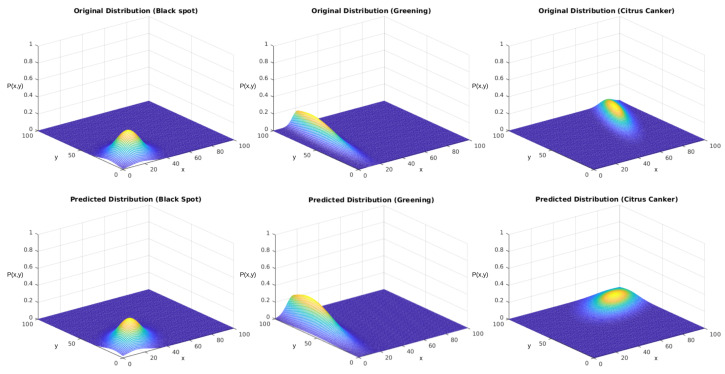
Predictions of the complete PF using the evolutionary algorithm. The *x* and *y* axes represent the same coordinates represented in Figure 5, and the P(x,y) axis represents the values obtained from Equation (Equation 1).

**Table 1 sensors-23-02165-t001:** Results for the timing comparison tests. The tests show significant statistical separation for the results of both algorithms.

	Faster R-CNN (ms)	YOLO-V3 (ms)	*p*-Value
Image 1	2264 ± 56	78 ± 1	$p<2.2\times 10^{-16}$
Image 2	2059 ± 36	90 ± 1	$p<2.2\times 10^{-16}$
Image 3	2367 ± 40	75 ± 1	$p<2.2\times 10^{-16}$
Image 4	2110 ± 29	91 ± 1	$p<2.2\times 10^{-16}$
Image 5	2246 ± 35	52 ± 1	$p<2.2\times 10^{-16}$
Image 6	2115 ± 26	53 ± 1	$p<2.2\times 10^{-16}$
Image 7	2016 ± 34	54 ± 0.4	$p<2.2\times 10^{-16}$
Image 8	2114 ± 40	67 ± 1	$p<2.2\times 10^{-16}$

**Table 2 sensors-23-02165-t002:** Metrics for the MobileNetV2 model.

	Precision	Recall	F1-Score	Support
Citrus Canker	1.00	1.00	1.00	201
Fresh Oranges	1.00	1.00	1.00	388
Greening	1.00	1.00	1.00	369
Black Spot	1.00	1.00	1.00	206
**Macro Average**	1.00	1.00	1.00	1164
**Weighted Average**	1.00	1.00	1.00	1164
**Global Accuracy:**	100%			

**Table 3 sensors-23-02165-t003:** Metrics for the EfficientNetV2-B0 model.

	Precision	Recall	F1-Score	Support
Citrus Canker	1.00	1.00	1.00	201
Fresh Oranges	1.00	1.00	1.00	388
Greening	1.00	1.00	1.00	369
Black Spot	1.00	1.00	1.00	206
**Macro Average**	1.00	1.00	1.00	1164
**Weighted Average**	1.00	1.00	1.00	1164
**Global Accuracy:**	100%			

**Table 4 sensors-23-02165-t004:** Metrics for the NASNet-Mobile model.

	Precision	Recall	F1-Score	Support
Citrus Canker	0.98	0.91	0.95	201
Fresh Oranges	1.00	1.00	1.00	388
Greening	1.00	1.00	1.00	369
Black Spot	0.92	0.99	0.95	206
**Macro Average**	0.98	0.97	0.97	1164
**Weighted Average**	0.98	0.98	0.98	1164
**Global Accuracy:**	98%			

**Table 5 sensors-23-02165-t005:** Results for the time analysis from the classification models. The tests display significant statistical separation for the results of each pair of algorithms.

	First Model	Second Model	*p*-Value
	MobileNetV2	NASNet-Mobile	
**times (ms)**	30 ± 3	44 ± 2	$p<2.2\times 10^{-16}$
	MobileNetV2	EfficientNetV2-B0	
**times (ms)**	30 ± 3	34 ± 2	$p<2.2\times 10^{-16}$
	EfficientNetV2-B0	NASNet-Mobile	
**times (ms)**	34 ± 2	44 ± 2	$p<2.2\times 10^{-16}$

**Table 6 sensors-23-02165-t006:** Parameters of the probability functions used in the first test.

Disease	$D_{0}$	$\mu_{x}$	$\mu_{y}$	$\sigma_{x}$	$\sigma_{y}$
Black spot	0.3	20	20	10	10
Greening	0.2	20	80	5	30
Citrus canker	0.1	80	80	10	20

**Table 7 sensors-23-02165-t007:** Parameters of the probability density functions used in the second test.

Disease	$D_{0}$	$\mu_{x}$	$\mu_{y}$	$\sigma_{x}$	$\sigma_{y}$	ρ
Black spot	0.3	20	20	10	10	0.2
Greening	0.2	20	80	5	30	0.5
Citrus canker	0.1	80	80	10	20	0.8

**Table 8 sensors-23-02165-t008:** Results for the score and obtained parameter considering the first experiment.

Disease		$D_{0}$	$\mu_{x}$	$\mu_{y}$	$\sigma_{x}$	$\sigma_{y}$
Black spot score: 0.9457	Original values	0.3	20	20	10	10
	Predicted values	0.25	20.88	20.97	11.30	9.70
Greening Score: 0.9630	Original values	0.2	20	80	5	30
	Predicted values	0.20	19.74	80.38	4.92	28.78
Citrus canker score: 0.9262	Original values	0.1	80	80	10	20
	Predicted values	0.10	81.31	80.49	10.60	18.29

**Table 9 sensors-23-02165-t009:** Results for the scores and obtained parameters considering the second experiment.

Disease		$D_{0}$	$\mu_{x}$	$\mu_{y}$	$\sigma_{x}$	$\sigma_{y}$	ρ
Black spot score: 0.8983	Original values	0.3	20	20	10	10	0.2
	Predicted values	0.29	20.91	21.90	11.56	11.86	0.46
Greening score: 0.7998	Original values	0.2	20	80	5	30	0.5
	Predicted values	0.20	14.86	77.56	6.47	28.60	0.39
Citrus canker score: 0.7531	Original values	0.1	80	80	10	20	0.8
	Predicted values	0.10	81.79	82.30	15.93	18.29	0.47

## Data Availability

The partial or total data and the codes will be made available in the case of manuscript approval.

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
