# Peer review of "Using Mobile Edge AI to Detect and Map Diseases in Citrus Orchards"

_sensors, 2023, doi:10.3390/s23042165_

Round 1
Reviewer 1 Report
This paper analyzes the advantages and disadvantages of citrus recognition and disease detection algorithms, and proposes a method to draw the spatial distribution map of citrus orchard diseases by combining GPS., which is innovative. The following parts of the paper need to be improved:
1. Why choose YOLOv3 and FAST-RCN as the citrus recognition algorithm (line136-149)?
2. Can the 120 sample pictures obtained on the network meet the training needs?(lin154).
3. Is it better to add the hardware and software description of edge ai to the materials and methods? Corresponding to the thesis title.
4. Is it better to add risk prediction test for non-infected citrus?Can this idea highlight the innovation of this paper?
Author Response
Please, check the attached document.

Reviewer 2 Report
1. Introduction
The introduction includes sufficient background and relevant references.
To justify the need to develop an application to recognize diseases and their spread pattern among orchards, it is desirable to add information on the amount of damage caused by the main citrus diseases.
2. Theoretical References and Related Work
Either this section or the Introduction should review and comment on similar work affecting citrus plants.
For example:
- Smartphone assist deep neural network to detect the citrus diseases in Agri-informatics https://doi.org/10.1016/j.gltp.2021.10.004
- A Smart Mobile Diagnosis System for Citrus Diseases Based on Densely Connected Convolutional Networks DOI:10.1109/ACCESS.2019.2924973
It is desirable to indicate the main differences, disadvantages and positive aspects.
3. Materials and Methods
In this section, the methodology for the search and classification of citrus fruits in orchards is considered in some detail. However, it is desirable to give explanations or links why YOLO-V3 (rather than YOLO-V5, YOLO-V6 or YOLO-V7) and Faster R-CNN were originally chosen.
It is worth noting the theoretically well-considered issue of mapping detected diseases. In Figures 5,6,7,8, it is necessary to sign the axes and their dimensions. Perhaps figure 5 is redundant.
4. Results
The results are presented in expanded form, especially the section on the probabilistic spread of diseases in the garden is worth noting.
I would like to look at the simulation of the situation in the presence of more than one focus of the disease, as well as the simulation when introducing means of treatment and the application of methods to combat the considered diseases
In figures 11,12,14,15,16 it is necessary to sign the axes and their dimensions.
5. Conclusions and Discussions
It is desirable to give in the conclusions the numerical values of the efficiency parameters of the developed system and more fully describe the expected benefits from its application.
Author Response
Please, check the attached document.
